# AUGMENTING OFFLINE REINFORCEMENT LEARNING WITH STATE-ONLY INTERACTIONS

## ABSTRACT

Batch offline data have been shown considerably beneficial for reinforcement learning. Their benefit is further amplified by upsampling with generative models. In this paper, we consider a novel opportunity where interaction with environment is feasible, but only restricted to observations, i.e., *no reward* feedback is available. This setting is broadly applicable, as simulators or even real cyber-physical systems are often accessible, while in contrast reward is often difficult or expensive to obtain. As a result, the learner must make good sense of the offline data to synthesize an efficient scheme of querying the transition of state. Our method first leverages online interactions to generate high-return trajectories via conditional diffusion models. They are then blended with the original offline trajectories through a stitching algorithm, and the resulting augmented data can be applied generically to downstream reinforcement learners. Superior empirical performance is demonstrated over state-of-the-art data augmentation methods that are extended to utilize state-only interactions.

## 1 INTRODUCTION

Data augmentation has long been an effective approach that boosts the performance of learning algorithms. They have been particularly useful for neural networks whose regularization and generalization properties are much harder to characterize than shallow models. In reinforcement learning (RL), experience replay (Fedus et al., 2020; Mnih et al., 2015) can be considered as augmenting the latest experiences with the past ones when updating the model. An even more effective paradigm is offline RL (Levine et al., 2020), where a batch of previously collected trajectories are used to boost online RL. They can be used to augment the online update buffer, or to pre-train a model that is subsequently fine-tuned by online RL.

Offline RL may suffer from sub-optimal batch data because the behavior policy could be sub-optimal, and the data may not have sufficiently covered the environment, especially the high-rewarding regions. To address this issue, stitching approaches have been studied (Li et al., 2024). Model-based trajectory stitching (MBTS, Hepburn & Montana, 2024) augments the offline dataset by connecting low-reward trajectories with high-reward ones. This is shown in Figure 1b, where two dashed lines are added, connecting to a point that is closer to a bridge (higher value). However, both Li et al. (2024) and Hepburn & Montana (2024) only stitch *existing* trajectories while no novel high-reward ones are generated.

This issue appears solvable by generative models. As shown in Figure 1c, SynthER augments the set of *transitions* by training a diffusion model (Lu et al., 2023). However, it does not generate *trajectories* which would require auto-regressive models. In our experiment, directly doing so performs worse than SynthER which upsamples transitions, suggesting that as a data augmentation approach to RL, auto-regressively generating trajectories may accumulate high bias and produce low return.

To tackle this problem, we make a new observation that some limited form of interactions are helpful. Indeed, online *state-only* interaction with environment is feasible in many applications. For example:

- RL has been adopted to improve treatment policy of chronic disease by factoring in the delayed effects of treatment (Weltz et al., 2022). Many digital twins are available that simulate a patient's condition (Tardini et al., 2022). However, toxicity feedback (part of the reward) may exhibit marked

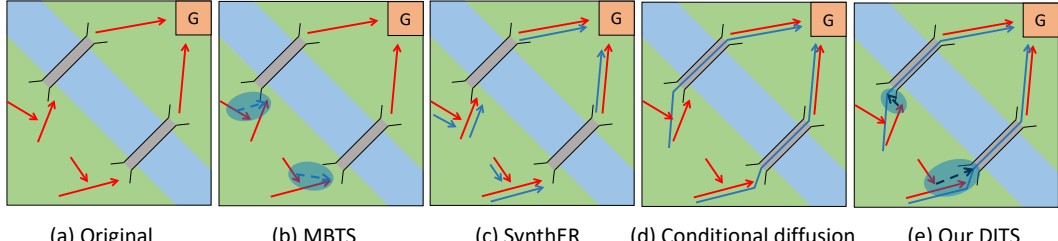

(a) Original     (b) MBTS     (c) SynthER     (d) Conditional diffusion     (e) Our DITS

Figure 1: An example that illustrates the difference between four data augmentation methods. **(a)**: the original offline dataset, where a river is in the main diagonal with two bridges. There are six random trajectories. Two of them walk to the Goal state (top right), which gives a high reward. Four trajectories are stuck at the lower-left half with a low reward. **(b)**: MBTS concatenates trajectories, but they are restricted to original trajectories, instead of leveraging the bridge-crossing one that could be synthesized by a generative model. **(c)**: the SynthER method where all transitions make an equal contribution to the training of the diffusion model. **(d)**: the conditional diffusion model which super-samples high-return trajectory regions. As a result, new trajectories that cross the bridges are formed. **(e)**: our DITS method of conditional diffusion followed by stitching, which concatenates low-return trajectories with high-return ones. In consequence, two dashed arrows are formed allowing two low-return trajectories to be connected to a bridge-crossing one.

- variance across patients, both physically and mentally. So it requires measurement on real persons from offline recordings.
- In inventory control, multi-product stock levels can often be well simulated for a stocking policy (Chen et al., 2024). However, the real reward (profit) also depends on the price which in turn fluctuates with the market and the marketing strategy (e.g., discount). So it is generally difficult to retrieve rewards in an online setting, and we need to best utilize the reward from the offline data.
- In sequential recommendation, it is common to model a user's behavior such as click-through rate. But the alignment with their ethical principle (reward) cannot be automatically inferred online (Stray et al., 2021).

The question we investigate and answer in the affirmative is: **can we improve trajectory generation, as a data augmentation approach for RL, by making *efficient* use of state-only interactions?**[1] A natural approach is to learn a good policy from offline RL, and deploy it by interacting with the environment to obtain a sequence of state and action. Finally, impute the reward from state and action via a regression model trained from offline data. Within this framework, our major contribution lies in two folds. Firstly, we make the key observation that although synthesizing high-return trajectories ameliorates the possible deficiency of such trajectories in the offline data, it may also lead to overfitting of such regions. Therefore, we blend them into the original dataset progressively with a *selective* stitching strategy, introducing new transitions between two states on *different* trajectories.

Secondly, we identify a policy model that best fits our setting. Since the interaction is not at test time, the policy's efficiency is not a main concern, allowing us to leverage Transformer or diffusion based planners. In particular, we extended the decision diffuser (DD, Ajay et al., 2023), which trains a diffusion model to generate trajectories *conditioned* on high return. For example, in Figure 1d, new trajectories are generated that cross the bridge to reach the goal state. Moreover, DD diffuses state sequences, which is exactly what is collected from new interactions. Such an alignment facilitates additional online fine-tuning of the diffuser; we leave it for future work.

We will refer to our method as Diffusion-based Trajectory Stitching (DITS). As shown in Figure 1e, DITS enjoys the benefit of conditional diffuser in generating new high-rewarding trajectories, and it also fixes the limitation of MBTS by allowing low-reward trajectories to connect with the high-reward ones generated by the diffuser. From the trajectory generation perspective, DITS circumvents auto-regression, delegating the backbone progression of state to direct interaction, reducing the task of trajectory generation to step-wise synthesis of actions and rewards. We will also show that both of them can be accomplished by DITS via a single diffusion model.

The workflow of DITS is also charted in Figure 2. The offline dataset is first used to train DITS' generator, producing trajectories that are blended with the original dataset through DITS' stitcher.

---

[1]We work on state interactions here, leaving the extension of observation-only interactions to future work.

Figure 2: **Di**ffusion-based **T**rajectory **S**titching (**DITS**). Trajectories from DITS' trajectory generator are combined with the original dataset for the stitching process, which creates new transitions (blue arrows in the "Trajectory stitching" block) and discards old transitions (grey dotted edges). Stitching was facilitated by DITS' reward and action generators. A filter is applied to prune away low-return trajectories, and the result is finally used by a downstream RL method.

The stitcher progressively evaluates the candidate states to transition to by using a series of criteria, leveraging the reward and actions generated by DITS. The resulting pool is filtered based on the full trajectory return, and the remaining trajectories are supplied to a downstream RL algorithm.

This paper is organized as follows. After the preliminaries in Section 2, we introduce the conditional decision diffuser used by DITS' generator, which produces full trajectories conditioned on high returns (Section 3). Based on it, the stitching algorithm is presented in Section 4, and its superior performance over other data augmentation methods is empirically demonstrated in Section 5.

**Related Work** Diffusion based models have been recently shown effective for RL. Diffusion policy models (Chi et al., 2023; Wang et al., 2023b; Ze et al., 2024) refine noise into actions through a gradient field. It is conditioned on the state or observation, and can be stably trained. The learned policy can accurately model multimodal and high-dimensional action distributions. In contrast, DD diffuses on state spaces instead of actions, and generates the action via an inverse dynamics model. It also allow more general conditioning such as high reward, constraint satisfaction, and skill composition. Wang et al. (2023a) proposed augmenting behavioral cloning by diffusion over state and action pairs, but it does not consider conditioning.

## 2 PRELIMINARY

We follow the standard setting of Markov decision process (MDP, Puterman, 1995), which is defined by the tuple $\langle \rho_0, \mathcal{S}, \mathcal{A}, \mathcal{T}, \mathcal{R}, \gamma \rangle$. Here $\rho_0$ is the initial distribution. $\mathcal{S}$ is the $m$-dimensional state space, and $\mathcal{A}$ is the action space. $\mathcal{T} : \mathcal{S} \times \mathcal{A} \to \Delta_{\mathcal{S}}$ is the transition function. $\mathcal{R} : \mathcal{S} \times \mathcal{A} \times \mathcal{S} \to \mathbb{R}$ is the reward function. $\gamma \in [0, 1)$ is a discount factor. The agent in this environment acts with a policy $\pi : \mathcal{S} \to \mathcal{A}$, generating a sequence of state-action-reward transitions, which represents a trajectory $\tau := (s_t, a_t, r_t)_{t \geq 0}$ with probability $p_\pi(\tau)$. The return of the trajectory is $R(\tau) := \sum_{t \geq 0} \gamma^t r_t$.

An offline dataset $\mathcal{D}$ consists of trajectories $(\tau_1, \ldots, \tau_T)$ which are collected using a behavior policy $\pi_\beta$. Each trajectory is composed of a set of transitions $\{(s_t, a_t, r_t, s'_t)\}$, where we customarily use $'$ to denote the *next* state, i.e., $s'_t = s_{t+1}$. SynthER (Lu et al., 2023) directly builds a generative model based on the union of such transition sets across all the $T$ trajectories to replay the experience.

**Diffusion Probabilistic Models (DDPM)** Diffusion models (Ho et al., 2020) are a class of generative models that learn the data distribution $q(\mathbf{x})$ from a dataset $\mathcal{D} := \{\mathbf{x}^i\}_{0 \leq i < M}$. One of the major usage of these models is generating high-quality images from text descriptions (Saharia et al., 2022). The diffusion models consist of two processes, which are the forward process and the reverse process. For the forward process, the model will add noise step by step via a predefined way $q(\mathbf{x}_{k+1}|\mathbf{x}_k) := \mathcal{N}(\mathbf{x}_{k+1}; \sqrt{\alpha_k}\mathbf{x}, (1 - \alpha_k)\mathbf{I})$. For the reverse process, the model has trainable parameters in order to learn a good way to denoise step by step, which can be represented mathematically as $p_\theta(\mathbf{x}_{k-1}|\mathbf{x}_k) := \mathcal{N}(\mathbf{x}_{k-1}|\mu_\theta(\mathbf{x}_k, k), \Sigma_k)$. In these processes, $\mathcal{N}(\mu, \Sigma)$ denotes a Gaussian distribution with mean $\mu$ and covariance matrix $\Sigma$, and $\alpha_k \in \mathbb{R}$ determines the variance schedule. $\mathbf{x}_0 := \mathbf{x}$ denotes the original sample, while $\mathbf{x}_1, \mathbf{x}_2, ..., \mathbf{x}_{K-1}$ are the intermediate steps for the diffusion, and $\mathbf{x}_K \sim \mathcal{N}(\mathbf{0}, \mathbf{I})$ is the unit Gaussian noise.

The reverse process model can be trained by minimizing the following loss (Ho et al., 2020):

$$\mathcal{L}_{denoise}(\theta) := \mathbb{E}_{k \sim \mathcal{U}[1,K], \mathbf{x}_0 \sim q, \epsilon \sim \mathcal{N}(\mathbf{0}, \mathbf{I})}[\|\epsilon - \epsilon_\theta(\mathbf{x}_k, k)\|^2].$$

The noise model $\epsilon_\theta(\mathbf{x}_k, k)$ is parameterized by a deep temporal U-Net (Janner et al., 2022). This is equivalent to modeling the mean of $p_\theta(\mathbf{x}_{k-1}|\mathbf{x}_k)$, as $\mu_\theta(x_{k-1}|x_k)$ can be calculated from $\epsilon_\theta(\mathbf{x}_k, k)$ (Ho et al., 2020).

**Guided Diffusion**   The diffusion training process can be guided using conditional data. In order to avoid the need of a strong classifier, classifier-free guidance method (Ho & Salimans, 2022) is preferred. This method learns both a conditional model $\epsilon_\theta(\mathbf{x}_k, \mathbf{y}, k)$ and an unconditional model $\epsilon_\theta(\mathbf{x}_k, \emptyset, k)$, where $\mathbf{y}$ is the label and $\emptyset$ is a dummy value that takes the place of $\mathbf{y}$ for unconditional situations. The perturbed noise can be denoted as $\epsilon_\theta(\mathbf{x}_k, \emptyset, k) + \omega(\epsilon_\theta(\mathbf{x}_k, \mathbf{y}, k) - \epsilon_\theta(\mathbf{x}_k, \emptyset, k))$, where $\omega$ denotes the guidance scale or the trade-off factor.

## 3   TRAJECTORY GENERATOR OF DITS WITH STATE-ONLY INTERACTION

We first present our trajectory generator, which produces $(s_t, a_t, r_t, s'_t)$ sequences through state-only interactions. A vanilla Decision Diffuser (DD, Ajay et al., 2023) works as a planner that proposes actions *conditioned* on high reward or other constraints, based on which the state-only interaction produces the next state $s'_t$. However, DD does not generate rewards, while many applications lack a reward formula, e.g., patient's mental stress level. We thus improve DD by also generating rewards.

Analogously to Ajay et al. (2023), we only diffuse on the states and rewards, because the sequence over actions, which are often represented as joint torques, tend to have higher frequency and are less smooth, making them much harder to model (Tedrake, 2022). We therefore decide to model the sequence of $(s_t, r_t, s'_t)$ by a diffusion model, and train an inverse dynamics model (IDM) to infer $a_t$ from $s_t$ and $s'_t$ (Section 3.4).

### 3.1   CONDITIONAL DIFFUSION MODELS FOR DECISION MAKING

We can formulate sequential decision-making as a problem of conditional generative modeling:

$$\max_\theta \ \mathbb{E}_{\tau \sim \mathcal{D}}[\log p_\theta(\mathbf{x}_0(\tau)|\mathbf{y}(\tau))]. \tag{1}$$

Our aim is to generate a *partial* trajectory $\mathbf{x}_0(\tau)$ using the information of conditioning data $\mathbf{y}(\tau)$. $\mathbf{x}_0(\tau)$ can be any subsequence of $\tau$, and $\mathbf{y}(\tau)$ can be the corresponding return, the constraints satisfied by it, or the skill demonstrated in it. Since DITS aims to generate high return trajectories, we will use the return $\sum_{i=t}^{t+H} \gamma^{i-t} r_i$ as our condition $\mathbf{y}(\tau)$, when $\vec{x}_0(\tau)$ assumes a subsequence of $\tau$ from $t$ to $t + H$. Following Ajay et al. (2023), we omit writing out the expectation over the random subsequence of $\tau$ in Equation 1.

We can then construct our generative model according to the conditional diffusion process:

$$q(\mathbf{x}_{k+1}(\tau)|\mathbf{x}_k(\tau)), \quad p_\theta(\mathbf{x}_{k-1}(\tau)|\mathbf{x}_k(\tau), \mathbf{y}(\tau)). \tag{2}$$

Here, $q$ represents the forward diffusion process while $p_\theta$ represents the trainable reverse process parameterized by $\theta$. Compared with the original Diffusion Probabilistic Model (Ho et al., 2020), the reverse process here has an additional conditioning label $\mathbf{y}(\tau)$, guiding the reverse process with high return condition and enabling the model to learn a good policy in the environment.

### 3.2   TRAJECTORY GENERATION IN DITS VIA STATE-ONLY INTERACTION

We will diffuse over the following state for diffusion timesteps $k = 1, \ldots, K$:

$$\mathbf{x}_k(\tau) \coloneqq (s_{t-C+1}, r_{t-C+1}, \ldots, s_t, r_t, \ldots, s_{t+H-C}, r_{t+H-C})_k \ \in \mathbb{R}^{H(m+1)}. \tag{3}$$

In the process of diffusion, it is crucial to make the current part of the trajectory consistent with the history. We therefore introduce the most recent $C$ number of states by repeatedly overwriting the entries of $s_{t-C+1}, \ldots, s_t$ in $\mathbf{x}_k(\tau)$, throughout all diffusion steps $k$. We finally extract $r_t$ and $s_{t+1}$ from $\mathbf{x}_0(\tau)$. If $s_{t+1}$ is used directly to run the denoising process again, we end up with an auto-regressive generation of trajectories, which turned out ineffective in our experiments.

DD trains an IDM to infer the action $a_t$ from $s_t$ and $s_{t+1}$, which can be used to interact with the environment and to obtain the true next state $s_{t+1}$. The whole generation procedure is formalized in Algorithm 1 and is illustrated in Figure 3. We will refer to the entry of $s_{t-C+1}$ as $\mathbf{x}_k(\tau)[0]$.state, and the entry of $r_{t-C+1}$ as $\mathbf{x}_k(\tau)[0]$.reward.

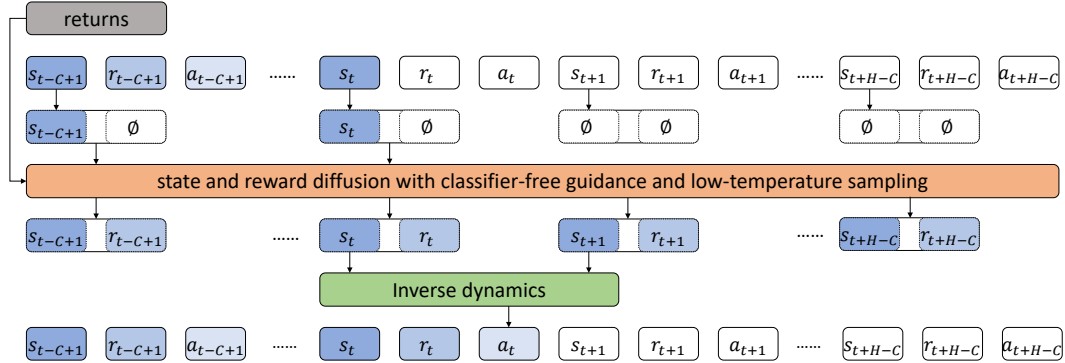

Figure 3: **Trajectory generation of DITS.** Given the latest $C$ number of states $s_{t-C+1}, \ldots, s_t$, the diffuser uses classifier-free guidance with low-temperature sampling to generate a sequence of future states and rewards. IDM is applied to generate action $a_t$ with $s_t$ and $s_{t+1}$. $\emptyset$ means the value is left for the diffuser to fill in, instead of clamping with the observation (if available).

---

**Algorithm 1:** Trajectory generation in DITS via state-only interaction with a *receding-horizon* control

1 **Input**: Noise model $\epsilon_\theta$, inverse dynamics $f_\phi$, guidance scale $\omega$, history length $C$, condition $\mathbf{y}$.
2 Initialize $h \leftarrow Queue(length = C)$. Initialize state $s_0$ and an empty trajectory $\tau$.
3 **for** $t = 0, 1, \ldots$ *until done* **do**
4      $h.insert(s_t)$. Initialize $\mathbf{x}_K(\tau) \sim \mathcal{N}(\mathbf{0}, \alpha\mathbf{I})$ in $\mathbb{R}^{H(m+1)}$ by Equation 3.
5      **for** $k = K \ldots 1$ **do**
6          $\mathbf{x}_k(\tau)[: length(h)]$.states $\leftarrow h$. Set $\hat{\epsilon}$ with Equation 4, using $\theta$, $\mathbf{x}_k(\tau)$, $\mathbf{y}$, and $k$.
7          Sample $\mathbf{x}_{k-1}(\tau) \sim \mathcal{N}(\mu_{k-1}, \alpha\Sigma_{k-1})$, where $(\mu_{k-1}, \Sigma_{k-1}) \leftarrow$ Denoise$(\mathbf{x}_k(\tau), \hat{\epsilon})$.
8      $r_t \leftarrow \mathbf{x}_0(\tau)[C-1]$.reward.    $s_{t+1} \leftarrow \mathbf{x}_0(\tau)[C]$.state.    $a_t \leftarrow f_\phi(s_t, s_{t+1})$ from IDM.
9      Apply $a_t$ to the environment. Overwrite $s_{t+1}$ with the true next state. $\tau$.append$(s_t, a_t, r_t, s_{t+1})$.

---

It is noteworthy that we do not write in the reward from the latest $C$ steps, differing from the states. Instead, we just let the diffuser fill in their values, which is consistent with our postulation that past states provide sufficient information to predict the reward. Further, although we do not pursue it in this work, our generator could also be used in online model-based planning, in which case the past rewards may not be even available to the agent online.

### 3.3 CONDITIONING WITH CLASSIFIER-FREE GUIDANCE AND THE TRAINING OBJECTIVE

To train the generator, we employ classifier-free guidance to integrate conditional influence into the diffusion process (Ho & Salimans, 2022). This is superior to using a classifier that requires estimating the $Q$-function (Ajay et al., 2023). It defines a perturbed noise that is applied at trajectory generation:

$$\hat{\epsilon} := \epsilon_\theta(\mathbf{x}_k(\tau), \emptyset, k) + \omega \cdot [\epsilon_\theta(\mathbf{x}_k(\tau), \mathbf{y}(\tau), k) - \epsilon_\theta(\mathbf{x}_k(\tau), \emptyset, k)], \quad (4)$$

where the scalar $\omega$ tends to augment and extract the best portions of trajectories in the dataset that comply with the conditioning of $\mathbf{y}(\tau)$, i.e., high return here. This helps the decision diffuser to learn a good policy from an average dataset. Then the reverse process $p_\theta$, which is parameterized by the noise model $\epsilon_\theta$, can be learned by minimizing (Ho & Salimans, 2022)

$$\mathcal{L}_{\text{gen}}(\theta) := \mathbb{E}_{\epsilon \sim \mathcal{N}(\mathbf{0}, \mathbf{I}), k \sim \mathcal{U}\{1, \ldots, K\}, \tau \in \mathcal{D}, \beta \sim \text{Bern}(p)}[\|\epsilon - \epsilon_\theta(\mathbf{x}_k(\tau), (1 - \beta)\mathbf{y}(\tau) + \beta\emptyset, k)\|^2]. \quad (5)$$

### 3.4 INVERSE DYNAMICS MODELS (IDMs)

Sampling states and rewards using a diffusion model is not enough for generating a full trajectory or extracting policy. A policy can be inferred from estimating the action $a_t$ that leads the state $s_t$ to $s_{t+1}$. Hepburn & Montana (2024) proposed implementing the IDM $p(a_t|s_t, s_{t+1})$ with a conditional variational autoencoder. However, they are generally harder to train with a more complicated structure than MLPs. We hence adopt the IDM from Pathak et al. (2018), which is also employed by DD and is shown to be effective than jointly diffusing over actions and states (Ajay et al., 2023). Let $a_t := f_\phi(s_t, s_{t+1})$, where $f_\phi$ is an MLP with parameter $\phi$. The training objective of IDM is simply

$$\mathcal{L}_{\text{IDM}}(\phi) := \mathbb{E}_{(s, a, s') \in \mathcal{D}}[\|a - f_\phi(s, s')\|^2]. \quad (6)$$

---

**Algorithm 2:** The stitching algorithm of DITS in brief. A detailed version is in Algorithm 4.

---

1   **Input:** the original offline dataset $\mathcal{D}_0$ with $T$ trajectories $(\tau_1, ..., \tau_T)$
2   **for** $k = 0, ..., K$ **do**
3      Generate $n$ trajectories $\mathcal{D}'_k$ by DITS' trajectory generator in Algorithm 1.
4      Set $\mathcal{D}_k \leftarrow \mathcal{D}_k \cup \mathcal{D}'_k$. Train the state-value function $V$ on dataset $\mathcal{D}_k$.
5      **for** $t = 1, ..., T$ *(i.e., length of $\mathcal{D}_k$)* **do**
6          $(s, s') \leftarrow (s_0, s'_0)$ from $\tau_t$. Initialize a new trajectory $\hat{\tau}_t$
7          **while** *not done (e.g., maximum length of episode not reached)* **do**
8              **if** *a state $\hat{s}'$ in $\mathcal{D}_k$ (possibly in a different trajectory than t) is **good** to stitch to* **then**
9                  Generate action $\tilde{a}$ based on $(s, \hat{s}')$ by DITS' action generator (e.g., IDM).
10                 Generate reward $\tilde{r}$ given $(s, \hat{s}', \tilde{a})$ by DITS' reward generator from Algorithm 3.
11                 Add $(s, \tilde{a}, \tilde{r}, \hat{s}')$ to the new trajectory $\hat{\tau}_t$
12                 $s \leftarrow \hat{s}'$, $s' \leftarrow$ the next state on the trajectory of $\hat{s}'$
13              **else**
14                 Add original transition to $\hat{\tau}_t$ and slide $(s, s')$
15      $\mathcal{D}_{k+1} \leftarrow T$ trajectories in $\{\hat{\tau}_t\}$ with the highest return
16   **Return:** dataset $\mathcal{D}_{K+1}$ for behavioural cloning training

---

**Algorithm 3:** Reward generation in DITS

---

1   **Input:** Noise model $\epsilon_\theta$, guidance scale $\omega$, condition $\mathbf{y}$, current state $s$, stitching candidate $s'$
2   Initialize $\mathbf{x}_K(\tau) \sim \mathcal{N}(\mathbf{0}, \alpha\mathbf{I})$ in $\mathbb{R}^{H(m+1)}$ from Equation 3
3   **for** $k = K...1$ **do**
4      $\mathbf{x}_k(\tau)[0]$.state $\leftarrow s$.    $\mathbf{x}_k(\tau)[1]$.state $\leftarrow s'$.    Set $\hat{\epsilon}$ with Equation 4, using $\theta, \mathbf{x}_k(\tau), \mathbf{y}$, and $k$.
5      Sample $\mathbf{x}_{k-1}(\tau) \sim \mathcal{N}(\mu_{k-1}, \alpha\Sigma_{k-1})$, where $(\mu_{k-1}, \Sigma_{k-1}) \leftarrow \text{Denoise}(\mathbf{x}_k(\tau), \hat{\epsilon})$.
6   **Return:** $\mathbf{x}_0(\tau)[0]$.reward

---

## 4   STITCHER OF DITS

As illustrated in Figure 1, merely generating high-return trajectories is sub-optimal as it may overfit those regions. We overcome this limitation by blending such trajectories into the original dataset progressively with a *selective* stitching strategy, thereby diversifying the high-return trajectories by, e.g., starting from a low-reward region.

Stitching algorithms have been studied in the literature. The Model-based Trajectory Stitching method (MBTS, Hepburn & Montana, 2024) augments the dataset $\mathcal{D}$ by stitching together high-value regions of different trajectories, i.e., introducing new transitions between two states on two different trajectories. The candidate states for stitching are determined by i) the state-value function $V(s)$ and ii) the forward dynamics probability $p(s'|s)$. However, MBTS does not generate new trajectories conditioned with high return, leaving the stitched data still sub-optimal.

To address this issue, we propose a stitching algorithm for DITS, with an abridged version shown in Algorithm 2, and a detailed version deferred to Algorithm 4 in Appendix A.1. It can be compared with MBTS which is recapped as Algorithm 5 in Appendix A.2. The key differences are highlighted in color, and are summarized in Appendix A.2.

The key step of Algorithm 2 is step 8, where a candidate state $\hat{s}'$ is evaluated for stitching. Here, $\hat{\cdot}$ stands for a candidate state. By "good", we follow MBTS and impose three conditions as detailed in Algorithm 4. Firstly, it needs to be close to $s'$, and we detail the neighborhood selection in Appendix B.1. Moreover, there is a good chance to switch to $\hat{s}'$, and the value of $\hat{s}'$ is higher than $s'$. We will describe these two conditions in Section 4.1 and Appendix A.4, respectively.

Whenever a stitching step occurs, the following key query is required in step 10 of Algorithm 4:

> As we transition from state $s$ to state $\hat{s}'$, how much would the reward $r$ be?

Since the state transition $s \rightarrow \hat{s}'$ is not present in the data, we need to complement it with reward. A straightforward solution is to reuse the trajectory diffuser by clamping the recent two states to $s$ and $\hat{s}'$. The details are given in Algorithm 3. We also tried to train a separate diffusion model for this purpose, and got similar results. The details are deferred to Appendix A.3.

Table 1: Comparison of **D4RL score** between DITS and **interaction-free** augmentation methods, including SynthER and MBTS. In general, our DITS shows significant improvement. The base offline RL methods are BC and TD3+BC. We ran 5 random seeds.

| | BC | | | | TD3+BC | | | |
|---|---|---|---|---|---|---|---|---|
| Methods | Original | SynthER | MBTS | **DITS** | Original | SynthER | MBTS | **DITS** |
| **Med-Expert** | | | | | | | | |
| Halfcheetah | 55.2±8.3 | 61.9±7.8 | **86.9**±**2.5** | 86.8±0.7 | 90.8±7.0 | 85.9±8.2 | **93.8**±**3.4** | 91.2±3.7 |
| Hopper | 52.5±7.7 | 61.2±8.8 | 94.8±11.7 | **105.1**±**6.1** | 101.1±10.5 | 102.5±10.9 | 109.1±3.9 | **111.3**±**3.6** |
| Walker2d | 107.5±2.8 | 108.2±3.3 | 108.8±5.5 | **109.1**±**0.1** | 110.0±0.4 | 110.1±0.3 | 110.3±0.4 | **110.4**±**0.8** |
| **Average** | 71.7±6.3 | 77.1±6.7 | 96.8±6.6 | **100.3**±**2.3** | 100.6±6.0 | 99.5±6.5 | **104.4**±**2.6** | 104.3±2.7 |
| **Medium** | | | | | | | | |
| Halfcheetah | 42.6±3.1 | 42.8±2.3 | 43.2±0.3 | **44.1**±**0.7** | 48.1±0.2 | 48.8±0.3 | 48.4±0.4 | **50.3**±**3.4** |
| Hopper | 52.9±2.4 | 58.1±5.7 | 64.3±4.2 | **83.2**±**8.1** | 60.4±4.0 | 63.0±4.3 | 64.1±4.4 | **76.6**±**6.4** |
| Walker2d | 75.3±6.2 | 74.7±5.5 | **78.8**±**1.2** | 78.5±2.0 | 82.7±5.5 | **85.2**±**1.1** | 84.2±1.4 | 83.3±3.6 |
| **Average** | 56.9±3.9 | 58.5±4.5 | 62.1±1.9 | **68.6**±**3.6** | 63.7±3.2 | 65.7±1.9 | 65.6±2.1 | **70.2**±**4.5** |
| **Med-Replay** | | | | | | | | |
| Halfcheetah | 36.6 ±3.1 | 34.1±2.5 | **39.8**±**0.6** | 38.9±3.2 | 44.6±3.3 | 44.7±0.6 | 43.8±0.5 | **45.1**±**5.5** |
| Hopper | 18.1±7.4 | 27.5±10.2 | 50.2±17.2 | **96.4**±**6.8** | 60.9±5.2 | 63.2±4.4 | 77.4±17.0 | **99.4**±**1.9** |
| Walker2d | 26.0±5.9 | 33.1±10.8 | 61.5±5.6 | **74.5**±**6.5** | 81.8±5.3 | 82.1±5.6 | 82.8±3.4 | **83.4**±**4.2** |
| **Average** | 26.9±5.5 | 31.6±7.8 | 50.5±7.8 | **69.9**±**5.5** | 62.4±4.6 | 63.3±3.5 | 68.0±7.0 | **76.0**±**3.9** |
| **Kitchen** | | | | | | | | |
| Mixed | 51.5±7.3 | 53.7±5.8 | 49.3±6.6 | **60.0**±**4.8** | 0.0±0.0 | 0.0±0.0 | 4.3±2.2 | **42.2**±**6.3** |
| Partial | 38.0±5.3 | 44.2±5.5 | 39.4±7.8 | **44.4**±**3.9** | 0.7±0.4 | 0.5±0.2 | 2.5±0.8 | **37.4**±**7.1** |
| **Average** | 44.8±6.3 | 49.0±5.7 | 44.4±7.2 | **52.2**±**4.4** | 0.4±0.2 | 0.3±0.1 | 3.4±1.5 | **39.8**±**6.7** |
| **Antmaze** | | | | | | | | |
| Umaze | 55.3±4.2 | 66.5±4.4 | 73.6±7.3 | **82.3**±**10.2** | 70.8±39.2 | 87.3±6.6 | 75.6±13.8 | **92.3**±**7.7** |
| U-Diverse | 47.3±4.1 | 58.4±3.8 | 63.8±4.8 | **65.5**±**8.8** | 44.8±11.6 | 57.8±7.6 | 59.7±6.3 | **65.2**±**6.3** |
| **Average** | 51.3±4.2 | 62.5±4.1 | 68.7±6.1 | **73.9**±**9.5** | 57.8±25.4 | 72.6±7.1 | 67.7±10.1 | **78.8**±**7.0** |

### 4.1 FORWARD DYNAMICS CRITERION

To determine whether a state can be a candidate next state for stitching, we need a criterion based on forward dynamics model $p(\hat{s}'|s)$, depicting the transition probability from state $s$ to the possible candidate state $\hat{s}'$, which may *differ* from the observed next state $s'$. We model the environment dynamics as a Gaussian distribution, which is common for continuous state-space applications (Janner et al., 2019). We create an ensemble $\mathcal{E}$ of $N$ dynamics models parameterized by $\xi^i$: $\{\hat{p}_{\xi^i}(s_{t+1}|s_t) = \mathcal{N}(\mu_{\xi^i}(s_t), \Sigma_{\xi^i}(s_t))\}_{i=1}^N$. Each model is trained via maximum likelihood estimation from the dataset $\mathcal{D}$, for which the loss can be formulated as:

$$\mathcal{L}_{\hat{p}}(\xi) := \mathbb{E}_{s,s'\sim\mathcal{D}}[(\mu_\xi(s) - s')^T \Sigma_\xi^{-1}(s)(\mu_\xi(s) - s') + \log |\Sigma_\xi(s)|], \tag{7}$$

where $|\cdot|$ is the determinant of a matrix. We train the ensemble using different parameter initializations for each model in order to take the epistemic uncertainty into account. The criterion for determining the candidate next state can be formulated in a conservative manner:

$$\min_{i\in\mathcal{E}} \hat{p}_{\xi^i}(\hat{s}'|s) > \text{mean}_{i\in\mathcal{E}} \hat{p}_{\xi^i}(s'|s). \tag{8}$$

## 5 EXPERIMENTS

We now demonstrate the empirical performance of DITS for offline RL with state-only interactions. We emphasize that our goal is data augmentation, instead of proposing a new RL algorithm that possibly intertwines with or extends another existing RL algorithm delicately. Therefore, the comparison is against other augmentation methods under a common set of base RL algorithms.

Table 2: Comparison of **D4RL score** between DITS and **interaction-based** augmentation methods, including the vanilla SynthER and its modified version. In general, our DITS shows significant improvement. The base offline RL method is **TD3+BC**. We ran 5 random seeds.

|  |  | SynthER (original) | SynthER (modified) | DITS |
|---|---|---|---|---|
| Med-Expert | Halfcheetah | 85.9±8.2 | 88.3±4.1 | **91.2±3.7** |
| Med-Expert | Hopper | 102.5±10.9 | 101.9±5.7 | **111.3±3.6** |
| Med-Expert | Walker2d | 110.1±0.3 | 110.2±0.5 | **110.4±0.8** |
| Medium | Halfcheetah | 48.4±0.3 | 47.3±1.5 | **50.3±3.4** |
| Medium | Hopper | 63.0±4.3 | 65.8±2.9 | **76.6±6.4** |
| Medium | Walker2d | 85.2±1.1 | **85.4±2.8** | 83.3±3.6 |
| Med-Replay | Halfcheetah | 44.7±0.6 | 44.0±0.9 | **45.1±5.5** |
| Med-Replay | Hopper | 63.2±4.4 | 68.5±2.8 | **99.4±1.9** |
| Med-Replay | Walker2d | 82.1±5.6 | **84.1±5.3** | 83.4±4.2 |

**Environments**   Although the three applications listed in the introduction are realistic in practice, we do not have access to their offline data. As a workaround, we employed standard locomotion tasks from the D4RL benchmark (Fu et al., 2020), including Halfcheetah, Hopper, and Walker2d. To explore its robustness, we further tested on Kitchen and Antmaze, two harder environments. We only considered the medium-expert, medium, and medium-replay datasets for locomotion tasks, because the downstream RL models already have excellent performance in expert datasets.

**Baseline data augmentation methods**   We first compared with state-of-the-art interaction-free data augmentation methods, including SynthER (Lu et al., 2023) and MBTS (Hepburn & Montana, 2024). Then we compared with an extended version of SynthER that utilizes state-only interactions. In particular, as SynthER generates $(s, a, r, s')$, we replaced the diffusion generated next state $s'$ with the actual next state retrieved from interaction. We kept the same number of generated samples between modified SynthER and DITS.

**Settings**   We evaluated the performance using the *normalized average return* (Fu et al., 2020). It first applies the offline learned policy to 10 online episodes. The total reward of each episode is referred to as a *score*, which is then normalized via $100 \times \frac{\text{score} - \text{random score}}{\text{expert score} - \text{random score}}$. The process is repeated with 5 random seeds, producing mean and variance. DITS used 300 trajectories generated from the DITS trajectory generator. The detailed hyperparameter settings are relegated to Appendix B.

## 5.1   Performance of normalized average return

The results of normalized average score for comparisons with *interaction-free* methods are shown in Table 1, where the base RL learners are behavioral cloning (BC) and TD3+BC (Fujimoto & Gu, 2021). A similar comparison with *interaction-based* methods is shown in Table 2, where we only compared with the modified SynthER on TD3+BC, because BC does not utilize the next state $s'$,

Our DITS made significant improvement on the Kitchen tasks for TD3+BC. The original TD3+BC performs poorly in this situation, along with the other augmentation methods under comparison. However, DITS is able to generate full demonstrative trajectories with good policies using conditional diffusion based on state-only interactions, resulting in a dramatic boost for the TD3+BC's performance. The improvement offered by DITS is more significant when the original dataset is not of high quality. Comparing the two columns of SynthER in Table 2, it does get improved by state-only interactions.

For a broader range of base RL methods, we evaluated our DITS augmentation based on CQL (Kumar et al., 2020), IQL (Kostrikov et al., 2021), and CPED (Zhang et al., 2024). Table 3 presents the performance on Hopper and Walker2d. Our DITS is generally able to improve the performance, especially on datasets with lower quality.

**Trajectory generation time cost**   We evaluated the time consumption of trajectory generation for DITS. Using Hopper Medium Expert as an example, the mean and standard deviation in seconds over 10 repeated trials are: 236.89±5.93 (200 steps), 474.69±11.24 (400 steps), 711.63±17.08 (600 steps), 950.03±21.85 (800 steps), 1185.19±23.83 (1000 steps). We ran on a single RTX3080 GPU.

Table 3: Improvement of D4RL score as DITS is applied in conjunction with CQL, IQL, and CPED.

| | | CQL | DITS+CQL | IQL | DITS+IQL | CPED | DITS+CPED |
|---|---|---|---|---|---|---|---|
| Hopper | Medium-Expert | 105.4 | **109.3±2.8** | 91.5 | **111.2±2.1** | 95.3±13.5 | **109.4±2.1** |
| Hopper | Medium | 58.5 | **77.2±7.5** | 66.3 | **77.9±5.5** | 100.1±2.8 | 97.9±4.9 |
| Hopper | Medium-Replay | 95.0 | **96.8±4.7** | 94.7 | **101.9±1.3** | 98.1±2.1 | **99.3±1.8** |
| Walker2d | Medium-Expert | **108.8** | **108.3±1.5** | 108.8 | **111.1±1.3** | 113.0±1.4 | 110.8±2.2 |
| Walker2d | Medium | 72.5 | **79.1±2.9** | 78.3 | **84.1±3.3** | 90.2±1.7 | **93.4±2.0** |
| Walker2d | Medium-Replay | 77.2 | **79.0±3.8** | 73.9 | **87.9±4.1** | 91.9±0.9 | **92.1±1.1** |

Table 4: **Correlation analysis.** While SynthER reconstructs the data from the original training set, we show that our method can improve the quality of the data with a higher similarity to the expert data. We used 100k samples from each model to compute the average result.

| | SynthER | | DITS | |
|---|---|---|---|---|
| | Marginal | Correlation | Marginal | Correlation |
| Hopper Med-Replay | **0.983** | **0.997** | 0.932 | 0.989 |
| Hopper Med-Expert | 0.958 | 0.992 | **0.982** | **0.995** |
| Hopper Expert | 0.934 | 0.982 | **0.998** | **0.998** |

## 5.2 CORRELATION ANALYSIS

Following Lu et al. (2023), we next measured the similarity between a) transitions generated SynthER and DITS diffusion model (without stitching) after training from the Hopper *Medium Replay* dataset, and b) samples from the *Medium Replay*, *Medium Expert* and *Expert* datasets. The metrics included the Kolmogorov-Smirnov statistic and the Pearson rank correlation (Patki et al., 2016). As shown in Table 4, DITS can generate data with the distribution close to the expert dataset while training only on medium replay data thanks to the conditional guidance. We refer to the KS test results as **Marginal**, and the Pearson statistics results as **Correlation**.

## 5.3 ABLATION STUDY ON THE DITS METHOD

Our first ablation study investigates how the two key ingredients of our method, the trajectory generator and the stitcher, contribute to its performance. Using BC as the base RL algorithm, We compared DITS with two variants: a) only adding DITS generated trajectories to the datasets based on state-only interactions, and b) only performing stitching on the original dataset without generating new trajectories.

The resulting normalized average score is presented in Figure 4, using the Hopper dataset and the same three difficulty levels as in Table 1. The results for Halfcheetah and Walker2d are deferred to Figure 6 in Appendix D. Overall, fully applying DITS is more effective than only utilizing the stitcher, which indicates that the conditional diffuser and interaction are indeed helpful by providing high-return trajectories. Furthermore, full DITS outperforms the method with stitcher removed, because the stitching step can avoid overfitting to the high-return regions by blending in transitions from the low value regions.

**Ablation on Conditional Guidance Scale** We also analyzed the sensitivity of the conditional guidance scale $\omega$ of the conditional diffusion model. On the Hopper Medium Expert dataset, we evaluated the generation quality by computing the correlation between expert data. The correlation turns out $0.996, 0.996, 0.993, 0.989$ for $\omega = 1.2, 1.4, 1.6, 1.8$, respectively.

**Ablation on the Number of Generated Trajectories** We next study the impact of the high-return trajectories generated by DITS via interaction. Figure 5 shows that, using Hopper medium expert as the environment and BC as the base offline RL algorithm, the performance of our method can be improved with more sampled trajectories. Further, the variance also decays as a broader range of situations are fed to BC. When only a small number of high-return trajectories are available, some test episodes turned out to suffer a considerably low score.

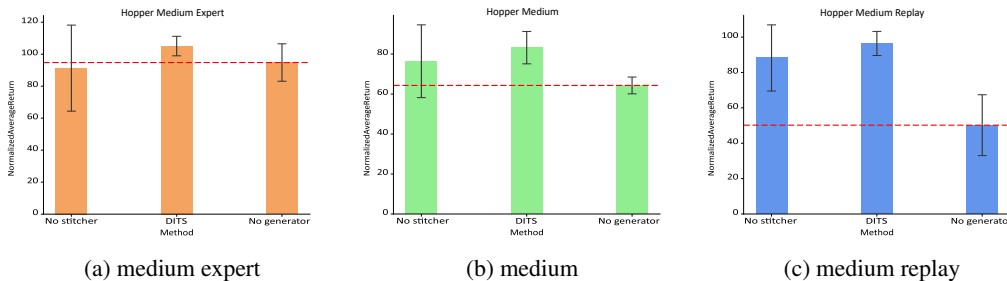

| (a) medium expert | (b) medium | (c) medium replay |
|---|---|---|

Figure 4: **Ablation study on trajectory generation and stitching**. We show the contribution of the two components of DITS by comparing its *normalized average return* on Hopper with two variants: no stitcher (dropping the stitching step) and no generator (dotted line, using the original dataset).

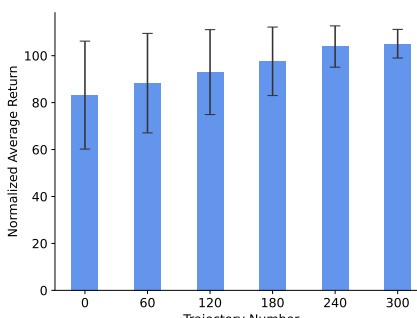

Figure 5: Ablation on the number of DITS generated trajectory

|  |  | DiffStitch | DITS |
|---|---|---|---|
| Med-Expert | Halfcheetah | **96.0±0.5** | 91.2±3.7 |
| Med-Expert | Hopper | 107.1±7.0 | **111.3±3.6** |
| Med-Expert | Walker2d | **110.2±0.3** | **110.4±0.8** |
| Medium | Halfcheetah | **50.4±0.5** | 50.3±3.4 |
| Medium | Hopper | 60.3±4.9 | **76.6±6.4** |
| Medium | Walker2d | **83.4±4.7** | 83.3±3.6 |
| Med-Replay | Halfcheetah | 44.7±0.3 | **45.1±5.5** |
| Med-Replay | Hopper | 79.6±13.5 | **99.4±1.9** |
| Med-Replay | Walker2d | **89.7±4.2** | 83.4±4.2 |

Table 5: Comparing DITS and DiffStitch in D4RL score

### 5.4 COMPARISON WITH DIFFSTITCH

DiffStitch (Li et al., 2024) is a recent work that uses diffusion imaginations to stitch trajectories that augment the data. However, it does not utilize state-only interactions, nor employ conditional guidance. Although we also use diffusion models, we use them to generate high-reward trajectories as candidates for stitching, instead of using them to implement the stitching method itself.

For completeness we compare in Table 5 the D4RL scores of our DITS and DiffStitch. The base RL method is TD3+BC, and 5 random seeds were used. Our DITS matches or outperforms DiffStitch in D4RL locomotion datasets.

Due to space limitation, we defer to Appendix E the accuracy of reward generation.

### 6 CONCLUSION, LIMITATION, BROADER IMPACT, AND FUTURE WORK

We proposed a novel data augmentation method DITS for offline RL, where state-only interactions are available with the environment. The generator based on conditional diffusion models allows high-return trajectories to be sampled, and the stitching algorithm blends them with the original ones. The resulting augmented dataset is shown to significantly boost the performance of base RL methods.

As a limitation, we did not update DITS' trajectory generator after new state-only sequences are obtained from the interaction. Future work will empower diffusion models to learn from partially observed data (no reward available). As for broader impact, DITS will benefit a range of applications where state-only interactions are available, e.g., healthcare, recommendation, and inventory control.

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

# A ALGORITHM DETAILS

## A.1 DIFFUSION-BASED TRAJECTORY STITCHING (DITS)

We present in Algorithm 4 the detailed pseudo-code for the stitching algorithm of DITS.

---

**Algorithm 4:** The stitching algorithm of DITS (detailed version). An abridged version is available in Algorithm 2.

---

1 **Input:** An inverse dynamics model $f_\phi$, a DITS trajectory generator $D_\theta$, an ensemble of dynamics models $\{\hat{p}_\xi^i(s_{t+1}|s_t)\}_{i=1}^N$, an acceptance threshold $\tilde{p}$, and a dataset $\mathcal{D}_0$ made up of $T$ trajectories $(\tau_1, ..., \tau_T)$, additional trajectory number $n$, sum of rewards threshold $\lambda$.

2 **for** $k = 0, ..., K$ **do**

3      Generate $n$ new trajectories $\mathcal{D}'_k$ using $D_\theta$ model with Algorithm 1

4      $\mathcal{D}_k \leftarrow concat(\mathcal{D}_k, \mathcal{D}'_k)$

5      Train state-value function $V$ on dataset $\mathcal{D}_k$.

6      **for** $t = 1, ..., length(\mathcal{D}_k)$ **do**

7          Select $(s, s') = (s_0, s'_0)$ from $\tau_t$

8          Initialize new trajectory $\hat{\tau}_t$

9          **while** *not done* **do**

10             Create a set of neighbourhood $\{\hat{s}_j\} = \mathcal{N}(s') \cap \{$states in $\mathcal{D}_k\}$

11             Let $j = \arg\max_i V(\hat{s}'_i)$

12             **if** $\min_i \hat{p}_\xi^i(\hat{s}'_j|s) > mean_i \hat{p}_\xi^i(s'|s)$ *and* $V(\hat{s}'_j) > V(s')$ **then**

13                Generate new action $\tilde{a} \sim f_\phi(s, \hat{s}'_j)$

14                Generate new reward $\tilde{r} \sim D_\theta(s, \hat{s}'_j)$ via Algorithm 3

15                Add $(s, \tilde{a}, \tilde{r}, \hat{s}'_j)$ to new trajectory $\hat{\tau}_t$

16                Set $s = \hat{s}'_j$, and $s'$ to the next state on the trajectory of $\hat{s}'_j$

17             **else**

18                Add original transition, $(s, a, r, s')$ to the new trajectory $\hat{\tau}_t$

19                Set $s = s'$, and $s'$ to the next state on the trajectory of $s$

20          **if** $\sum_{i\in\hat{\tau}_t} r_i \leq (1 + \tilde{p}) \sum_{i\in\tau_t} r_i$ **then**

21             $\hat{\tau}_t = \tau_t$

22      Sort the trajectories with the values of $\sum_{i\in\hat{\tau}_t} r_i$

23      $\mathcal{D}_{k+1} \leftarrow$ the top $T$ trajectories from the sorting result

24 $\mathcal{D}_{Aug} \leftarrow$ Select the trajectories $\tau$ in the dataset $\mathcal{D}_{k+1}$ satisfying $\sum_{i\in\tau} r_i > \lambda$

25 Train downstream RL models on dataset $\mathcal{D}_{Aug}$

---

## A.2 MODEL-BASED TRAJECTORY STITCHING (MBTS)

We next recap, in Algorithm 5, the Model-based Trajectory Stitching (MBTS) method from (Hepburn & Montana, 2024).

---

**Algorithm 5:** Model-based Trajectory Stitching (MBTS)

1   **Input:** a dataset $\mathcal{D}_0$ with $T$ trajectories $(\tau_1, ..., \tau_T)$
2   **for** $k = 0, ..., K$ **do**
3      Train the state-value function $V$ on dataset $\mathcal{D}_k$.
4      **for** $t = 1, ..., T$ *(i.e., length of $\mathcal{D}_k$)* **do**
5          $(s, s') \leftarrow (s_0, s'_0)$ from $\tau_t$.
6          Initialize a new trajectory $\hat{\tau}_t$
7          **while** *not done* **do**
8             **if** *a state $\hat{s}'$ in $\mathcal{D}_k$ (possibly in a different trajectory than t) is close to $s'$ and is good to stitch to* **then**
9                Generate action $\tilde{a}$ based on $(s, \hat{s}')$ by IDM
10               Generate reward $\tilde{r}$ under $(s, \hat{s}', \tilde{a})$ by WGAN
11               Add $(s, \tilde{a}, \tilde{r}, \hat{s}')$ to the new trajectory $\hat{\tau}_t$
12               $s \leftarrow \hat{s}'$
13               $s' \leftarrow$ the next state on the trajectory of $\hat{s}'$
14             **else**
15               Add original transition to $\hat{\tau}_t$ and slide $(s, s')$
16      $\mathcal{D}_{k+1} \leftarrow (\hat{\tau}_1, \ldots, \hat{\tau}_T)$           $\triangleright$ $\mathcal{D}_k$ has constant length $T$

---

## A.3 GENERATING REWARD BY TRAINING A SEPARATE DIFFUSER

We also tried to impute the reward by learning another diffusion model, defining the states as

$$\mathbf{x}_k(\tau) := (s_t, s_{t+1}, r_t, s_{t+1}, s_{t+2}, r_{t+1}, \ldots, s_{t+H-1}, s_{t+H}, r_{t+H-1})_k \in \mathbb{R}^{H(2m+1)}, \quad (9)$$

for $k \in [1, K]$. During the diffusion process, we introduce the given $s_t$ and $s_{t+1}$ by clamping the first two entries of $\mathbf{x}_k(\tau)$ for all diffusion steps $k$. We do not enforce that the forth generated entry must be equal to the second one as in Equation 9, although they are obviously equal at training phase. It is also possible to take into account the state of the past $C$ number of steps. However, in practice we noticed that this does not provide much improvement, suggesting that $s_t$ and $s_{t+1}$ are sufficient to predict $r_t$. Both the training and generation processes are similar to those of the trajectory generator, and the latter process is detailed in Algorithm 6.

---

**Algorithm 6:** Reward generation by using a separate diffuser with states in Equation 9

1   **Input:** Noise model $\epsilon_\theta$, guidance scale $\omega$, condition $\mathbf{y}$, current state $s$, stitching candidate $s'$
2   Initialize $\mathbf{x}_K(\tau) \sim \mathcal{N}(\mathbf{0}, \alpha\mathbf{I})$ in $\mathbb{R}^{H(2m+1)}$ from Equation 9
3   **for** $k = K...1$ **do**
4      $\mathbf{x}_k(\tau)[0]$.states $\leftarrow (s, s')$. Set $\hat{\epsilon}$ with Equation 4, using $\theta$, $\mathbf{x}_k(\tau)$, $\mathbf{y}$, and $k$.
5      Sample $\mathbf{x}_{k-1}(\tau) \sim \mathcal{N}(\mu_{k-1}, \alpha\Sigma_{k-1})$, where $(\mu_{k-1}, \Sigma_{k-1}) \leftarrow$ Denoise$(\mathbf{x}_k(\tau), \hat{\epsilon})$.
6   **Return:** $\mathbf{x}_0(\tau)[0]$.reward, i.e., the $r_t$ in Equation 9

---

## A.4 VALUE FUNCTION CRITERION

In order to measure if a state is worth stitching, a common approach trains a value function over $\mathcal{D}$. As the states on the DDR-generated trajectories tend to have a high value, they enjoy a higher chance of stitching, hence promoting these states to be blended into the original dataset.

We approximate the value function using an MLP with parameter $\beta$. The Bellman error on $\mathcal{D}$ is

$$\mathcal{L}_V(\beta) := \mathbb{E}_{s,r,s' \sim \mathcal{D}}[(r + \gamma V_\beta(s') - V_\beta(s))^2]. \quad (10)$$

## B  HYPERPARAMETER AND ARCHITECTURAL DETAILS

In this section, we describe the hyperparameter choice for our model and the architectural details for our model:

- We use a temporal U-Net (Janner et al., 2022) for the noise $\epsilon_\theta$ modeling. It consists a U-Net structure with 6 repeated residual blocks, while each block consisting two temporal convolutions, each followed by group norm (Wu & He, 2018), and a final Mish nonlinearity (Misra, 2019). Both of the timestep and condition embeddings are 128-dimensional vectors produced by two separate 2-layered MLP with 256 hidden units and Mish nonlinearity. The embeddings are concatenated together before getting added to the activations of the first temporal convolution within each block. Our code for the DDR model is a modification of the code for the original Decision Diffuser (Ajay et al., 2023), for which the link is `https://github.com/anuragajay/decision-diffuser`.
- We use a 2-layered MLP with 512 hidden units and ReLU activations for the modeling of the inverse dynamics model $f_\phi$.
- We train $\epsilon_\theta$ and $f_\phi$ using the Adam optimizer (Kingma & Ba, 2015) with a learning rate $2 \cdot 10^{-4}$ and batch size of 32 for $1e6$ training steps.
- We choose the probability $p$ of removing the conditioning information to be 0.25.
- We use $K = 200$ diffusion steps.
- We choose context length $C \in \{1, 20\}$, $C = 20$ is preferred in the Kitchen datasets. Both $C$ values are able to generate decent results in the locomotion datasets, but $C = 20$ tends to have more stability.
- We use a planning horizon $H$ with $H - C = 100$ in all the D4RL locomotion tasks, while using $H - C = 56$ in D4RL kitchen tasks.
- We use a guidance scale $\omega \in \{1.2, 1.4, 1.6, 1.8\}$ but the exact choice varies by task.
- We choose $\alpha = 0.5$ for low temperature sampling.
- We use a 3-layered MLP with 300 hidden units with ReLU activation to model the foward dynamics. The network takes a state $s$ for input and output a mean $\mu$ and a standard deviation $\sigma$ for a Gaussian distribution $\mathcal{N}(\mu, \sigma^2)$. For all experiments, an ensemble size of 7 is used with the best 3 being chosen. We train the forward dynamics models with Adam optimizer (Kingma & Ba, 2015), a learning rate of $3 \cdot 10^{-4}$ and a batch size of 256. We initialize the parameter with different random seeds for each forward dynamics model in the ensemble.
- We choose to use a 2-layered MLP with 256 hidden units and a ReLU activation to parameterize the value function. We train two value functions with different parameter initialization and take the minimum of the two during the stitching process. We train the model with Adam optimizer and a learning rate of $3 \cdot 10^{-4}$ and a batch size of 256.
- We choose the neighbourhood radius $\rho \in \{0.1, 1.0, 3.0\}$ while the exact choice varies by task.
- We use the additional trajectory number $n = 300$ for each epoch during stitching.
- We use the sum of rewards threshold $\lambda = \max_t(\sum_{\tau_t} r^i) - \kappa$, while $\kappa$ is in the range of $[500, 1000]$ depending on the task.
- We choose the acceptance threshold $\tilde{p} = 0.1$ to ensure the stitched trajectory only to be used when a significant improvement is guaranteed.
- We use a behavioural cloning model parameterized by a 2-layered MLP with 256 hidden units and ReLU activation.
- We choose to use the epoch number $K = 1, 2, 3, 4$. For most cases, using $K = 2$ will have the sufficient performance gain, and the results will saturate in the following epochs.

### B.1  NEIGHBORHOOD SELECTION

For trajectory stitching, we discourage stitching two states that are far away. In this case, stitching in a created neighbourhood for the next state is preferred. Multiple metrics are available to define the

neighbourhood by measuring the distance in the state space. Castro (2020) proposed a pseudo-metric, but its computational cost is rather high. Therefore, we resort to a more straightforward approach by applying the L2 norm for neighbourhood selection:

$$\mathcal{N}(s) := \{\hat{s} : \|\hat{s} - s\|_2 < \rho\}, \tag{11}$$

where $\rho$ is a hyperparameter with a relatively small value.

## C  CLASSIFIER-FREE GUIDANCE DETAILS

In this section, we show the derivation of the classifier-free guidance Eq.4 for completeness. From the derivation outlined in prior works (Luo, 2022), we know that $\nabla_{\mathbf{x}_k(\tau)} \log q(\mathbf{x}_k(\tau)|\mathbf{y}(\tau)) \propto -\epsilon_\theta(\mathbf{x}_k(\tau), \mathbf{y}(\tau), k)$. Furthermore, we can derive:

$$q(\mathbf{x}_k(\tau)|\mathbf{y}(\tau)) = q(\mathbf{x}_k(\tau)) \frac{q(\mathbf{x}_k(\tau)|\mathbf{y}(\tau))}{q(\mathbf{x}_k(\tau))}$$

$$\Rightarrow \log q(\mathbf{x}_k(\tau)|\mathbf{y}(\tau)) = \log q(\mathbf{x}_k(\tau)) + (\log q(\mathbf{x}_k(\tau)|\mathbf{y}(\tau)) - \log q(\mathbf{x}_k(\tau)))$$

In order to sample from $q(\mathbf{x}_0(\tau)|\mathbf{y}(\tau))$ with classifier-free guidance, we multiply the second term with conditional guidance factor $\omega$:

$$\log \hat{q} := \log q(\mathbf{x}_k(\tau)) + \omega(\log q(\mathbf{x}_k(\tau)|\mathbf{y}(\tau)) - \log q(\mathbf{x}_k(\tau)))$$

$$\Rightarrow \nabla_{\mathbf{x}_k(\tau)} \log \hat{q} = \nabla_{\mathbf{x}_k(\tau)} \log q(\mathbf{x}_k(\tau)) + \omega(\nabla_{\mathbf{x}_k(\tau)} \log q(\mathbf{x}_k(\tau)|\mathbf{y}(\tau)) - \nabla_{\mathbf{x}_k(\tau)} \log q(\mathbf{x}_k(\tau)))$$

$$\Rightarrow \hat{\epsilon} := \epsilon_\theta(\mathbf{x}_k(\tau), \emptyset, k) + \omega(\epsilon_\theta(\mathbf{x}_k(\tau), \mathbf{y}(\tau), k) - \epsilon_\theta(\mathbf{x}_k(\tau), \emptyset, k))$$

In this case, we have our Eq.4. This result can also be expanded to a composing form with a number of different conditioning variables (Ajay et al., 2023).

## D  ADDITIONAL ABLATIONS OF DITS COMPONENTS

### D.1  ABLATION ON GENERATION AND STITCHING

We performed ablation studies on the Halfcheetah and Walker2d environments, in addition to the ablation study in Section 5.3. The settings are the same as there, and the results are shown in Figure 6. The observations are similar to those in Section 5.3.

## E  ASSESSMENT OF REWARD PREDICTION

In this section, we analyze the reward generation ability of the DITS model and compare our model with an MLP generator and WGAN generator to show that our model has a better reward generating performance. We evaluate the mean MSE error of the predicted reward and the true reward over the evaluating batch $\mathcal{D}_B$ via the metric:

$$M = \mathbb{E}_{s,s',r \sim \mathcal{D}_B}(D_\theta(s, s') - r)^2 \tag{12}$$

We compare our generator with 2-layered MLP generator with 256 hidden units and a WGAN with 2-dimensional latent space and 256 hidden units on the D4RL Hopper Medium-Expert dataset. The results are shown in Figure 7.

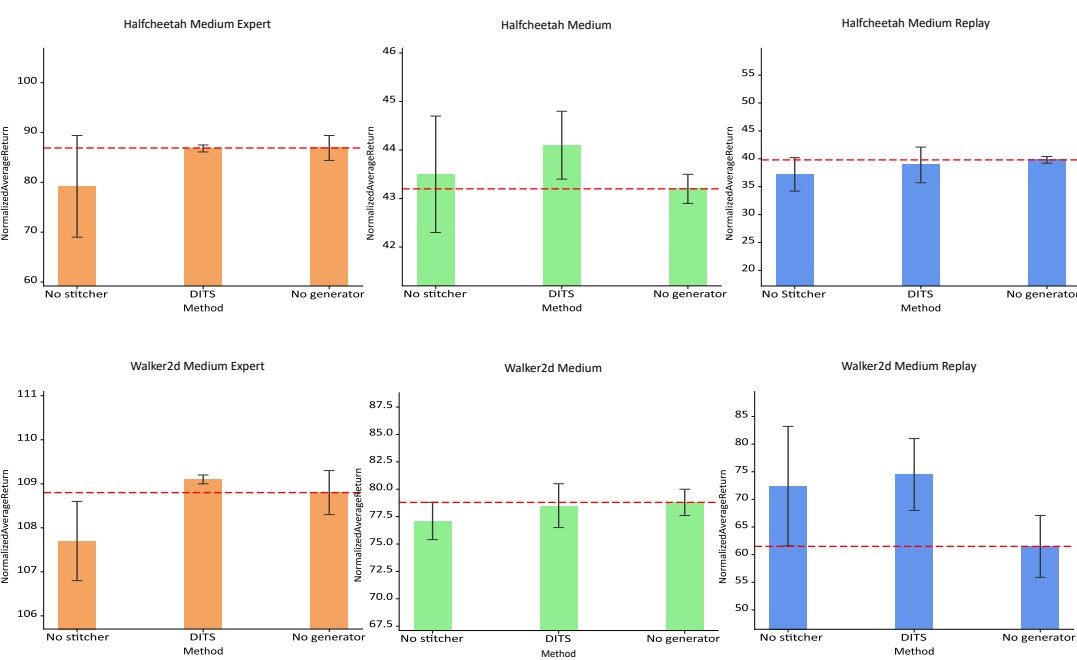

Figure 6: **Ablation study on trajectory generation and stitching**. The setting is the same as Figure 4, except that the datasets are Halfcheetah (top) and Walker2d (bottom). Difficulty levels are medium expert, medium, and medium replay (left to right).

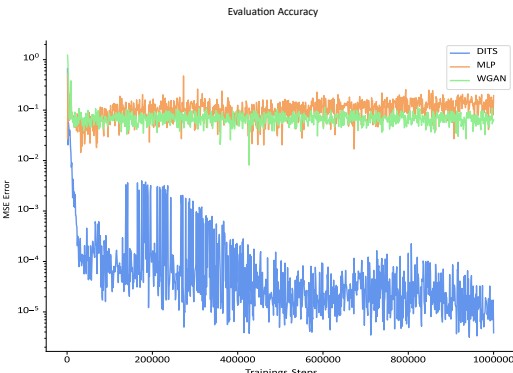

Figure 7: **The reward generation accuracy analysis**. We trained the DITS trajectory generation model on the Hopper Medium Expert dataset and evaluated the MSE error of the DITS model's generation on the environment during the training process. We compared our results with an MLP generator and a WGAN generator.

