# OpenReview forum: "Augmenting Offline Reinforcement Learning with State-only Interactions"
_ICLR.cc/2025/Conference — ICLR 2025 Conference Withdrawn Submission_

### Official Review · Reviewer_Awxi · 2024-11-04

**Soundness:** 3
**Presentation:** 1
**Contribution:** 2
**Rating:** 3
**Confidence:** 4

**Summary:**

This paper proposes a data augmentation strategy DITS to investigate the following question: can we improve trajectory generation, as a data augmentation approach for RL, by making efficient use of state-only interactions?

**Strengths:**

1. The goal of generating high-return trajectories via augmentation is well-motivated and worthwhile to the research community. I expand upon this more below.
1. Empirical setup is adequate. I think it's very appropriate to use behavior cloning (BC) as one of the evaluated algorithms, since DITS aims to generate high-return trajectories. If DITS performs well with BC, then that directly supports this claim.
1. Figure 1 does a nice job illustrating how DITS differs from previous approaches while also highlighting the advantage of DITS.

**Weaknesses:**

I vote to reject primarily because DITS seems to outperform baselines in only half of the considered tasks.

* **DITS outperforms baselines in only about half of tasks.** Table 1 reports the average return +/- 1 standard deviation (68% confidence interval). DITS's confidence interval overlaps with the intervals of other baselines in about half of the tasks, indicating that DITS does outperforms baselines. For instance:
  * All Med-Expert experiments, both BC and TD3+BC
  * Medium HalfCheetah, both BC and TD3+BC
  * Med Replay HalfCheetah TD3+BC
  * Kitchen Partial, both BC and TD3+BC
  * Antmaze U-Diverse both BC and TD3+BC, Umaze BC

* **Standard errors are not propagated properly.** When averaging performance across multiple experiments, the standard deviations should be added in quadrature and not simply averaged [1].

* **Standard deviation is likely not the greatest measure of uncertainty for this empirical analysis, since it assumes the underlying distribution of returns is normally distributed -- which is often not the case in RL.** A bootstrap confidence interval is more appropriate, as it does not assume the returns follow a particular distribution [2]

* **The related work is minimal and excludes a number of relevant data augmentation works.** Corrado et. al [3] focuses on generating expert quality augmented data for offline learning, similar to what DITS aims to do. Pitis et. al [5] is another model-based data augmentation technique, and Sinha et. al [4] consider offline RL when evaluating their augmentation strategy. I also list a few other data augmentation works [6-9] that can provide readers with a more comprehensive overview of the data augmentation research landscape and better contextualize DITS.

* "merely generating high-return trajectories is sub-optimal as it may overfit those regions" This statement seems plausible, though the paper does not provide evidence to support this claim. Kumar et. al [10] provide a theoretical argument that may support it; offline RL performs better with noisy expert data than purely expert data, indicating that high-return data may not be sufficient.

[1] Taylor, John R. An Introduction to Error Analysis: The Study of Uncertainties in Physical Measurements. 1997.

[2] Agarwal et. al. Deep Reinforcement Learning at the Edge of the Statistical Precipice. Neurips 2021. https://arxiv.org/abs/2108.13264

[3] Corrado et. al. Guided Data Augmentation for Offline Reinforcement Learning and Imitation Learning. https://arxiv.org/abs/2310.18247

[4] Sinha et. al. S4RL: Surprisingly Simple Self-Supervision for Offline Reinforcement Learning in Robotics. https://arxiv.org/abs/2103.06326

[5] Pitis et. al. MoCoDA: Model-based Counterfactual Data Augmentation. NeurIPS 2022. https://arxiv.org/abs/2210.11287

[6] Corrado & Hanna. Understanding when Dynamics-Invariant Data Augmentations Benefit Model-free Reinforcement Learning Updates. ICLR 2024. https://arxiv.org/abs/2310.17786

[7] Pitis et. al. “Counterfactual Data Augmentation using Locally Factored Dynamics.” NeurIPS 2020. https://arxiv.org/abs/2007.02863

[8] Abdolhosseini et. al. “On Learning Symmetric Locomotion.” ACM SIGGRAPH 2019. https://www.cs.ubc.ca/~van/papers/2019-MIG-symmetry/index.html

[9] Adrychowicz et. al. "Hindsight Experience Replay." NeurIPS 2017. https://arxiv.org/abs/1707.01495

[10] Kumar et. al. When Should We Prefer Offline Reinforcement Learning Over Behavioral Cloning? ICLR 2022. https://arxiv.org/abs/2204.05618

**Questions:**

See weaknesses.

---

> ### Author Response · Authors · 2024-12-03
>
> We thank the reviewer for the constructive review.
>
> **Q1**: Questions regarding experimental results analysis.
>
> **A1**: We appreciate the reviewer’s suggestion to include standard deviations for the baseline results to enable more accurate comparisons. We will incorporate this in our future work.
>
> **Q2**: Incomplete related works.
>
> **A2**: We thank the reviewer for pointing out the additional related works. We will address this in our future revision.
>
> **Q3**: DITS seems to outperform baselines in only half of the considered tasks.
>
> **A3**: In RL experiments, the results are generally far more mixed than supervised learning, typically with high variance and flipped superiority on different environments. So outperforming in half of the tasks while not losing much in most of the rest is already quite a good result.

---

### Official Review · Reviewer_ahba · 2024-11-04

**Soundness:** 2
**Presentation:** 2
**Contribution:** 2
**Rating:** 3
**Confidence:** 4

**Summary:**

The paper proposes a method to use a diffusion model trained on offline trajectories alongside an inverse dynamics and one-step dynamics model to synthesize new trajectories of higher return from low value trajectories.

**Strengths:**

The proposed method performs well compared to baselines in D4RL.

**Weaknesses:**

1. The algorithm seems like a minor iterative improvement without significant novel contribution. The “state-only trajectory modeling” which the paper highlights alot has been used in several previous works, including Decision Diffuser which the authors acknowledge (and in fact use almost the same setup).
2. The only addition over decision diffuser is the new stitching, but this approach seems very problematic. For one, the new transition \hat{s’} being sampled from a different trajectory will almost always be much less likely than s’ in the predicted trajectory. This is simply because most states in most trajectories are likely very different than any given state compared a predicted next state.
3. The proposed method is also just a minor variation from DiffStitch and SynthER.
3. It is puzzling why the authors use much slower autoregressive generation of the trajectory with the diffusion model instead of joint trajectory diffusion. It is unclear if using a diffusion model here affords any advantage.
4. Even more evidence that a diffusion model is not required is that the one step dynamics model p(s’ | s) actually used to verify if stitching to some state is allowed are simply modeled as Gaussians anyway. Why not just these models and remove the diffusion models?

**Questions:**

1. In section 3.2, the authors claim the last C steps are needed to make the trajectory consistent. However the tasks in D4RL that are evaluated here are all Markovian. Additionally here the authors claim just using s_{t+1} to run the denoising procedure instead of also using the history would make it autoregressive decoding. However, the current process is also autoregressive decoding. It seems like the authors are stating conditioning on history is not autoregressive. I would like some clarification on all this.
2. Why use autoregressive generation to begin with? This seems very inefficient when the diffusion model can be directly used to jointly diffuse whole trajectories (similar to Diffuser).

---

> ### Author Response · Authors · 2024-12-03
>
> We thank the reviewer for the constructive review.
>
> **Q1-a**: In section 3.2, the authors claim the last C steps are needed to make the trajectory consistent. However the tasks in D4RL that are evaluated here are all Markovian.
>
> **A1-a**: While we agree that the underlying physics are generally first-order Markovian in D4RL (we will omit "first-order" henceforth), the behavior policy used to generate the offline data does not necessarily *only* look at $s_t$, i.e., not Markovian.  To quote the original D4RL paper:
>
> **AntMaze**: "the controllers for this task are non-Markovian as they rely on tracking visited waypoints."
>
> **Kitchen**: "we use 3 datasets of human demonstrations". Humans are generally not Markovian.
>
>
> As a result, it is unclear whether the offline RL algorithm should assume the state trajectory $\{s_t\}$ is Markovian. In **Mujoco**, the policy does depend only on $s_t$, but to quote the D4RL paper:
>
> > The “medium-replay” dataset consists of recording all samples in the replay buffer observed during training until the policy reaches the “medium” level of performance.
> >
> > we further introduce a “medium-expert” dataset by mixing equal amounts of expert demonstrations and suboptimal data, generated via a partially trained policy or by unrolling a uniform-at-random policy.
>
> So the offline data is generated by a mixture of (many) policies, which may *leave* the function space of our policy, even though the mixture remains Markovian. Using a function space with overly high capacity may suffer from overfitting.
>
>
> Even in *online* RL with a Markovian dynamics, looking further into the history can be beneficial for several reasons:
>
> 1. **Approximation Errors in Function Approximation**
> Most RL algorithms, particularly in large or continuous state spaces, rely on function approximators like neural networks. These approximators can struggle to perfectly capture the dynamics or the value function, hence benefitting from additional context, such as past states, to improve prediction accuracy and to learn richer representations of the environment.
>
> 2. **Exploration and Strategy**
> In some environments, the history can inform an agent about its past actions, helping it to avoid revisiting unproductive regions of the state space, and to identify exploration strategies (e.g., avoid loops or backtracking). For example, in sparse-reward environments, remembering past states can help the agent understand how far it has explored and where it should focus.
>
> 3. **Policy Regularization**
> Using history can act as a form of implicit regularization, smoothing the policy and making it less sensitive to small variations in the current state. This can stabilize learning, particularly in environments with high noise or chaotic dynamics.
>
> 4. **Long-Term Dependencies in Policy**
> For some tasks, even if the environment is Markovian, achieving the optimal behavior might require strategies that depend on recognizing long-term patterns. For instance: a robot navigating a maze might benefit from recalling previously visited dead ends. A trading algorithm may perform better if it considers historical trends in market behavior.
>
> 5. **Empirical Success**
> In many benchmarks, adding historical context (e.g., via recurrent architectures or stacking observations) has shown to empirically improve performance, even in tasks that are theoretically Markovian. This suggests that the additional capacity to model temporal dependencies compensates for other limitations in the agent's design or training process. For example, the Decision Diffuser [1] does exactly this.

---

> > ### Author Response · Authors · 2024-12-03
> >
> > **Q1-b**: Additionally here (Section 3.2) the authors claim just using $s_{t+1}$ to run the denoising procedure instead of also using the history would make it autoregressive decoding. However, the current process is also autoregressive decoding. It seems like the authors are stating conditioning on history is not autoregressive. I would like some clarification on all this.
> >
> > **A1-b**: We do not fully understand this question, but let us try to answer it with our best effort. Each time, we perform diffusion over a segment of the trajectory containing $H$ states and rewards. Inside the diffusion model, these $H$ steps are not processed with any temporal perspective - they are just concatenated as shown in Equation 3. The only output we use from the diffusion model is the next state $s_{t+1}$, based on which we retrieve the action by IDM and finally interact with the environment; see Algorithm 1.  In our experiment, we found that planning ahead (i.e., modeling up to $s_{t+H-C}$ instead of $s_{t+1}$) improves the accuracy of predicting $s_{t+1}$. The Decision Diffuser [1] also employs this method of diffusion generation.
> >
> > We stress that **our trajectory generation is not auto-regressive**. An auto-regressive model uses the model’s own prediction to make further predictions into the future. We do not.  In fact, we use "teacher forcing" [2, pp 372], which is a strategy for training recurrent models that uses **ground truth as input**, instead of model output from a prior time step as an input. This is also how Transformers are trained, while their inference is auto-regressive. In our case, the teacher is the state-only interaction with the environment, which provides the true next state.
> >
> > **Q2**: Why use auto-regressive generation to begin with? This seems very inefficient when the diffusion model can be directly used to jointly diffuse whole trajectories (similar to Diffuser).
> >
> > **A2**: Echoing Q1-b, we are **not** performing auto-regressive generation. Neither do Decision Diffuser [1] and Diffuser [3] - they diffuse a window to predict just the next action. Then they perform this action and observe the real next state, which is then used for planning the next step in a *teacher forcing* style.  We also adopt this principle.
> >
> >
> > As for why not "diffuse *whole* trajectories", we assume it refers to diffusing jointly over state+action+rewards. Based on the empirical findings in [1], the high-frequency nature of actions brings about poor performance if one diffuses over trajectories that include actions. Therefore, we follow [1] to diffuse only over states and rewards, while generating actions using an inverse dynamics model.
> >
> >
> >
> > **Q3**: The algorithm seems like a minor iterative improvement without significant novel contribution. The “state-only trajectory modeling” which the paper highlights a lot has been used in several previous works, including Decision Diffuser which the authors acknowledge (and in fact use almost the same setup).
> >
> > **A3**: We respectfully disagree with this comment.  It is  unfair to downplay our contribution by saying that “state-only trajectory modeling” is also used in Decision Diffuser (DD). Our paper addresses data augmentation that generically serves multiple downstream RL algorithms (as demonstrated in our experiments), while DD is a fixed planning algorithm.
> >
> > One of our key contributions is to identify that certain (test-time) planning algorithms provide useful trajectory modeling methods that can be used to augment the data for offline RL. **Now that offline RL has witnessed its recent renaissance for over 5 years, can you name any paper that leverages “state-only interactions" to augment the data for offline RL?  If not, then why is our work considered "a minor iterative improvement without significant novel contribution"**?  As it resonates with hindsight bias, everything seems trivial once you know it.
> >
> > **Q4**: The proposed method is also just a minor variation from DiffStitch and SynthER.
> >
> > **A4**: We once more respectfully disagree with the "minor variation" evaluation. Neither DiffStitch nor SynthER uses online state-only interactions. SynthER simply generates new step-wise transitions, and for a fair comparison, we have already considered a variation in Section 5.1 that replaces the generated new state with the result from the state-only interaction. Table 2 shows it underperforms to our DITS.  This is expected, because it does not condition on high reward, and it does not incorporate a short history in the past which can be helpful.
> >
> > It is not clear to us how to extend DiffStitch to leverage state-only interactions.

---

> > > ### Author Response · Authors · 2024-12-03
> > >
> > > **Q5**: Even more evidence that a diffusion model is not required is that the one step dynamics model p(s’ | s) actually used to verify if stitching to some state is allowed are simply modeled as Gaussians anyway. Why not just these models and remove the diffusion models?
> > >
> > > **A5**: We do this simplified approach in the stitching process entirely due to computational efficiency, because there are many candidate transitions under consideration and the selection algorithm has to be very efficient. However, diffusion models pose far less a challenge in computation for trajectory generation. So why not use it to enjoy the better performance?
> > >
> > >
> > > [1] Ajay, A., Du, Y., Gupta, A., Tenenbaum, J., Jaakkola, T., & Agrawal, P. (2023). Is conditional generative modeling all you need for decision-making? ICLR.
> > >
> > > [2] Goodfellow, I., Bengio, Y., Courville, A. (2016). Deep Learning. The MIT Press.
> > >
> > > [3] Janner, M, Du, Y., Tenenbaum, J., Levine, S. (2022). Planning with Diffusion for Flexible Behavior Synthesis. ICML.

---

### Official Review · Reviewer_dBGZ · 2024-11-04

**Soundness:** 2
**Presentation:** 2
**Contribution:** 3
**Rating:** 5
**Confidence:** 4

**Summary:**

This paper introduces a new diffusion-based data augmentation method named DITS. DITS considers the setting that only state interactions are available with the interaction. The generator based on conditional diffusion models allows high-return trajectories to be sampled, and the stitching algorithm blends them with the original ones.

**Strengths:**

1. The idea of combining diffusion-based trajectory generating and trajectory stitching is novel and interesting.
2. The writing of the paper is clear and easy to follow.

**Weaknesses:**

1. The main contribution of this paper can be divided into two aspects. One is to generate more accurate transitions based on the state-only observable environment. However, this part is more like directly using diffusion-based planning methods to do data augmentation, while remaining the rewards as predicted. It is doubtful that this scheme is useful enough as it requires too much interaction with the environment like online reinforcement learning although it doesn't need the reward. Also, an additional ablation study is needed to show the effectiveness of this generation pipeline like using Diffuser, DD, or other trajectory-level data augmentation methods to generate trajectories given that the methods can access the dynamics of the environment.

2. The second aspect is to blend the high-returned trajectories with the original trajectories. The current paper lacks an analysis of the importance of the trajectory stitcher.

3. The main results in Table 1. only contain baselines BC and TD3+BC, which ignore many state-of-the-art offline RL baselines including CQL, IQL, DT, etc.

4. The data augmentation baseline only includes SynthER as the diffusion-based data augmentation methods, while currently there are many latest diffusion-based data augmentation methods like MTDiff. Given the fact that SynthER is a transition-level generation method, the comparison is unfair and incomplete.

**Questions:**

1. In Line 205, What is the meaning of m?
2. What is the meaning of equation 8?
3. How does the iteration rounds K affect the performance?

---

> ### Author Response · Authors · 2024-12-03
>
> We thank the reviewer for the constructive review.
>
> **Q1**: In Line 205, What is the meaning of $m$?
>
> **A1**: $m$ denotes the dimensionality of the state space $\mathcal{S}$; please see line 140.
>
> **Q2**: What is the meaning of Equation 8?
>
> **A2**: This equation defines the criterion for establishing a new stitching connection. Intuitively, it compares the probability of transitioning to $\hat s'$ and to $s'$. But the former is based on the minimal value over the ensemble, while the latter uses its mean, creating a conservative and stringent condition of preferring the proposed $\hat s'$ to the original $s'$.
>
> **Q3**: How does the iteration rounds K affect the performance?
>
> **A3**: As the number of rounds $K$ increases, the performance of the downstream RL method improves but eventually saturates beyond a certain $K$. Empirically, we find that $K = 2$ is sufficient for most cases, as discussed in Appendix B.
>
> **Q4**: The second aspect is to blend the high-returned trajectories with the original trajectories. The current paper lacks an analysis of the importance of the trajectory stitcher.
>
> **A4**: We have done ablation study on it. Please refer to Section 5.3 and Figure 4. We also give a conceptual illustration in Figure 1(d).
>
> **Q5**: The main results in Table 1 only contain baselines BC and TD3+BC, which ignore many state-of-the-art offline RL baselines including CQL, IQL, DT, etc.
>
> **A5**: We have indeed compared with CQL, IQL, and CPED in Table 3, focusing on Walker2d and Hopper. We will extend the comparisons to other environments (Halfcheetah, Kitchen, Antmaze).
>
> **Q6**: The data augmentation baseline only includes SynthER as the diffusion-based data augmentation methods, while currently there are many latest diffusion-based data augmentation methods like MTDiff. Given the fact that SynthER is a transition-level generation method, the comparison is unfair and incomplete.
>
> **A6**: We will compare with MTDiff in the future. However, we are not sure how it can take advantage of state-only interactions. So we will only compare in its vanilla form.
>
> **Q7**: The main contribution of this paper can be divided into two aspects. One is to generate more accurate transitions based on the state-only observable environment. However, this part is more like directly using diffusion-based planning methods to do data augmentation, while remaining the rewards as predicted. It is doubtful that this scheme is useful enough as it requires too much interaction with the environment like online reinforcement learning although it doesn't need the reward.
>
> Also, an additional ablation study is needed to show the effectiveness of this generation pipeline like using Diffuser, DD, or other trajectory-level data augmentation methods to generate trajectories given that the methods can access the dynamics of the environment.
>
> **A7**: Our experiment used state-only interactions to generate only 300 trajectories (line 410). As a result, the number of state-only interactions is at most 30\% of the sample size in Mujoco and AntMaze, and 22\% in Kitchen. We also kept the same number of generated samples/transitions between modified SynthER and DITS (line 405).
>
>
>
> In the applications we motivated in Section 1, the state-only interactions can be obtained with ease, typically through a simulator. But the reward can be much harder to get. We understand many applications do not fall into this setting, but it is indeed a realistic application scenario.
>
> We will add the suggested ablation studies in the future revision.

---

### Official Review · Reviewer_ms28 · 2024-11-04

**Soundness:** 3
**Presentation:** 3
**Contribution:** 3
**Rating:** 5
**Confidence:** 4

**Summary:**

This paper introduces Diffusion-based Trajectory Stitching (DITS) to enhance offline reinforcement learning by using state-only interactions. DITS consists of two basic parts: a DD-based trajectory generator and a stitcher. First, the generator obtains state-only trajectories from the real environment, and the corresponding actions and rewards are filled by an inverse dynamics model and the diffusion model, respectively. Then, the stitcher finds possible $(s, s')$ to stitch them up, forming new trajectories. Finally, the dataset containing both real trajectories and stitched trajectories are used for the policy training. The experiments show performance gain of DITS over the underlying offline RL algorithms.

**Strengths:**

- The settings of this paper is quite practical. It addresses real-world scenarios where reward feedback is difficult to obtain, making the method broadly applicable to various fields like healthcare and inventory control.
- This paper contains intensive experiments that show the effectiveness and robustness of DITS, proving its potential for real-world application.

**Weaknesses:**

See questions.

**Questions:**

- In Figure 2, the trajectories from DITS trajectory generator are annotated as "trajectories from good policies". However, they are actually obtained by interacting with the real environment using a planning-like method. DITS uses the DD-based trajectory generator to plan the following states, and predicts the action by an IDM. I think this annotation is somehow misleading, and pointing out they are from the **real environment** might be better.
- In Line 16, Algo 2, as you train various offline RL methods besides BC on $\mathcal{D}_{K+1}$, it is imprecise to say "for behavioural cloning training".
- In Algo 4, how do you train the state-value function $V$? Is it trained from scratch for each iteration $k$, or only fine-tuned?
- For experiments, some of the ablation studies takes BC as the base RL algorithm. However, as DITS generates rewards for new transitions, and Appendix E proves the accuracy of the generated rewards, training BC is a little wasting because it does not utilize reward information. TD3+BC, a simple modification of BC, seems more likely to take full advantages of DITS. Could you please conduct ablation studies that comparing DITS against DITS w/o sitcher and DITS w/o generator, as well as the ablation on the number of generated trajectories, using TD3+BC as the base RL policy? If they cannot be completed due to the time limit, I will also appriciate it.

---

> ### Author Response · Authors · 2024-12-03
>
> We thank the reviewer for the constructive review.
>
> **Q1**: In Figure 2, the trajectories from DITS trajectory generator are annotated as "trajectories from good policies". However, they are actually obtained by interacting with the real environment using a planning-like method. DITS uses the DD-based trajectory generator to plan the following states, and predicts the action by an IDM. I think this annotation is somehow misleading, and pointing out they are from the real environment might be better.
>
> **A1**: We appreciate the reviewer's feedback and acknowledge that the statement could be misleading. We will revise this annotation into "trajectories from state-only interactions".
>
> **Q2**: In Line 16, Algo 2, as you train various offline RL methods besides BC on $\mathcal{D}_{k+1}$, it is imprecise to say "for behavioural cloning training".
>
> **A2**: We agree with the reviewer that this statement is misleading. We will revise it to: "for downstream RL method training."
>
> **Q3**: In Algo 4, how do you train the state-value function $V$? Is it trained from scratch for each iteration $k$, or only fine-tuned?
>
> **A3**: At each iteration $k$, we train the state-value function $V$ from scratch. As the value network is a lightweight three-layer MLP, the computational overhead of training remains low.
>
> **Q4**: For experiments, some of the ablation studies takes BC as the base RL algorithm. However, as DITS generates rewards for new transitions, and Appendix E proves the accuracy of the generated rewards, training BC is a little wasting because it does not utilize reward information. TD3+BC, a simple modification of BC, seems more likely to take full advantages of DITS. Could you please conduct ablation studies that comparing DITS against DITS w/o sitcher and DITS w/o generator, as well as the ablation on the number of generated trajectories, using TD3+BC as the base RL policy? If they cannot be completed due to the time limit, I will also appriciate it.
>
> **A4**: We agree with the reviewer that using TD3+BC as the baseline for the ablation studies is more appropriate. We will incorporate it in future work.

---

### Note · Authors · 2024-12-16

I have read and agree with the venue's withdrawal policy on behalf of myself and my co-authors.